# Known and Potential Invertebrate Vectors of Raspberry Viruses

**DOI:** 10.3390/v14030571

**Published:** 2022-03-10

**Authors:** Jiunn Luh Tan, Nina Trandem, Jana Fránová, Zhibo Hamborg, Dag-Ragnar Blystad, Rostislav Zemek

**Affiliations:** 1Department of Zoology, Faculty of Science, University of South Bohemia, 37005 České Budějovice, Czech Republic; 2Biology Centre CAS, Institute of Entomology, 37005 České Budějovice, Czech Republic; rosta@entu.cas.cz; 3Division of Biotechnology and Plant Health, Norwegian Institute of Bioeconomy Research (NIBIO), 1433 Ås, Norway; nina.trandem@nibio.no (N.T.); zhibo.hamborg@nibio.no (Z.H.); dag-ragnar.blystad@nibio.no (D.-R.B.); 4Biology Centre CAS, Institute of Plant Molecular Biology, 37005 České Budějovice, Czech Republic; jana@umbr.cas.cz

**Keywords:** *Rubus idaeus*, aphids, mites, nematodes, arthropod pests, soft fruit, integrated pest management, virus control, virus-vector interactions, virus transmission

## Abstract

The estimated global production of raspberry from year 2016 to 2020 averaged 846,515 tons. The most common cultivated *Rubus* spp. is European red raspberry (*Rubus idaeus* L. subsp. *idaeus*). Often cultivated for its high nutritional value, the red raspberry (*Rubus idaeus*) is susceptible to multiple viruses that lead to yield loss. These viruses are transmitted through different mechanisms, of which one is invertebrate vectors. Aphids and nematodes are known to be vectors of specific raspberry viruses. However, there are still other potential raspberry virus vectors that are not well-studied. This review aimed to provide an overview of studies related to this topic. All the known invertebrates feeding on raspberry were summarized. Eight species of aphids and seven species of plant-parasitic nematodes were the only proven raspberry virus vectors. In addition, the eriophyid mite, *Phyllocoptes gracilis*, has been suggested as the natural vector of raspberry leaf blotch virus based on the current available evidence. Interactions between vector and non-vector herbivore may promote the spread of raspberry viruses. As a conclusion, there are still multiple aspects of this topic that require further studies to get a better understanding of the interactions among the viral pathogens, invertebrate vectors, and non-vectors in the raspberry agroecosystem. Eventually, this will assist in development of better pest management strategies.

## 1. Introduction

Insects are the largest group of invertebrates that act as plant virus vectors. In addition, mites and nematodes are also common plant virus vectors [1]. There are four known modes of viral transmission by arthropod vectors (insects and mites): (1) non-persistent, (2) semi-persistent, (3) circulative persistent, and (4) propagative persistent. The non-persistent viruses are acquired from an infected host within a minute and inoculation into a healthy plant takes a few seconds or minutes. However, the retention of non-persistent viruses is limited to the arthropod’s stylet and is bound to last only a few minutes to hours, or to next molt. Likewise, the semi-persistent viruses require slightly longer acquisition time (minutes to several hours), but inoculation only takes a few seconds to minutes. The semi-persistent viruses have a longer retention period (up to days) than non-persistent ones because the virus also accumulates itself in the arthropod’s anterior gut instead of only the stylet. There are two types of persistent viruses, namely circulative and propagative. Both of them require much longer acquisition and inoculation time, which range from minutes to hours, and a latent period of up to weeks. The virus has a very long retention period, ranging from days to potentially the entire life span of the vector. The difference between circulative and propagative is that propagative viruses are capable of replicating in the vector while circulative cannot. Certain propagative viruses are even transmitted to the offspring from parents. The persistent viruses are highly vector specific as they need to traverse multiple barriers within the vector [2,3,4,5].

Nematodes are known to transmit soil-borne viruses from four genera, namely *Nepovirus* (family *Secoviridae*), *Tobravirus* (family *Virgaviridae*), *Cheravirus* (family *Secoviridae*), and *Sadwavirus* (family *Secoviridae*) [6,7,8,9]. Andret-Link and Fuchs [10] categorized the nematode mode of viral transmission as semi-persistent, but there are some differences from the arthropod’s semi-persistent mechanism. The nematodes require a minimum period (ranging from few minutes to hours) to acquire viruses from infected plants and they are able to retain them for months or years. Furthermore, virus particles are retained on the esophageal surface of the nematode during ingestion of plant cell contents and if it is not bound, the virus particles will be digested in the intestine and excreted [11]. However, the virus particles will be lost after molting, and are not passed on to, or retained in eggs [9]. This is because the virus is not associated with the nematode’s body tissue but is trapped or bound to the esophagus surface, which is shed or altered during molting [12].

The interaction between plant virus and its vector often exhibits a certain degree of specificity [10]. This is also the case of raspberry viruses. Several invertebrate pests, such as aphids and nematodes, have been proven as the vector of specific raspberry viruses. However, there are still other potential raspberry virus vectors that are not well-studied. Therefore, this review aims to provide an overview of current available studies related to this topic.

## 2. Raspberry

*Rubus* (family Rosaceae) is the genus that includes blackberry (*Rubus fruticosus* L.), raspberry (*Rubus idaeus* L.), and their hybrids. According to FAOSTAT [13], the global raspberry production has been increasing since 2015, and the five-year (2016–2020) average of the total global production was estimated at 846,515 tons. Raspberry can be consumed fresh or processed [14]. The increasing demand can be, at least partly, attributed to the numerous nutritional contents of these berries, such as essential minerals and vitamins. Furthermore, the raspberry also contains high levels of antioxidants, which potentially result in numerous health benefits [15]. Three species are the most commercially important, namely, the European red raspberry (*R. idaeus* L. subsp. *idaeus*), the North American red raspberry (*R. idaeus* subsp. *strigosus* (Michx.)) and the black raspberry (*Rubus occidentalis* L.) [16]. Among these, the European red raspberry is the most commonly cultivated [17]. Red raspberry is a soft fruit species of temperate shrub, which has woody shoots and a perennial root system. In general, there are two types of raspberry cultivars, namely biennial-fruiting and annual-fruiting [18]. The biennial-fruiting cultivars (also known as floricane fruiting) take two years to complete their life cycle, which involves vegetative growth, flower initiation and fruit development, and induction and breaking of bud dormancy. On the other hand, the annual-fruiting cultivars (also known as primocane fruiting) complete the cycle of vegetative growth, flowering, and fruiting within a single growing season [19]. In this cultivar type, flowering begins July and August, where the canes are still actively elongating and as a result flower development are initiated in late summer and early autumn. Thus, fruiting can occur in the first year of development, completing the cycle within a year. On the other hand, in reality, the annual-fruiting cultivars may be delayed and produce half the fruit in the first year and the rest of the fruit in the second year, while certain biennial fruiting cultivars may also flower earlier, thus producing a small amount of fruit during the first year [20]. This kind of cultivars is sometimes considered as the third type, exhibiting traits intermediate between the two general cultivar types and originating from either biennial- or annual-fruiting cultivars [21]. Selecting good cultivars is important as it will affect the yield and quality of the fruits. Garcia, et al. [22] highlighted that the external appearance and texture of raspberries, such as size, firmness, freshness, and damage resistance of fruits, can be critical quality features for the market value. Some raspberry cultivars, such as Glen Ample, Tulameen, Autumn Bliss, Autumn Britten, Polka, and Himbotop, are widely grown due to their plant growth performance and good fruit quality [23]. Furthermore, new innovative methods, particularly the use of plastic tunnels, have been introduced to increase the yield of raspberry by providing a more favorable growing environment. This means less wind, more diffuse light conditions and protection against rain and frost, which result in a longer harvest period [24,25,26]. Although plastic tunnels increase the yield, they have also brought notable changes in the pest complex. Two-spotted spider mites and whiteflies are more frequently found in tunnel-grown raspberries, although aphids, leafhoppers, and thrips are also common. In field-grown raspberries, other pests such as beetles, moths, and hemipteran bugs are more abundant [24,26]. Therefore, knowledge about the abundance of pests in different growing environments is vital in implementing effective pest management strategies.

## 3. Known Plant Viruses Infecting Raspberry

Raspberry plants are affected by various plant viruses from several families and genera. Currently, 22 viruses are known to infect raspberry (Table 1). They can be transmitted via several modes. Typically, a plant virus is transmitted from an infected plant to a healthy plant, but rarely through the direct contact between these plants. The transmission happens through a causing agent or event, such as invertebrates, mechanical injury, and propagation. The plant virus transmission can be categorized into two modes of transmission: (1) vertical transmission and (2) horizontal transmission. Vertical transmission happens when the virus is inherited by the progeny from an infected parent plant through seeds or pollens, and vegetative propagation. Whereas, horizontal transmission happens between individuals of the same generation through fungi, invertebrate vectors, and mechanical or sap inoculation [5,27].

## 4. Known and Potential Invertebrate Vectors of Raspberry Viruses

The most commonly known plant virus vectors are insects of the orders Hemiptera (aphids, whiteflies, leafhoppers, treehoppers, and plant bugs) and Thysanoptera (thrips), and mites of the families Eriophyidae, Tenuipalpidae, and Tetranychidae [4,60]. Plant-parasitic nematodes that are known to transmit plant viruses belong to the order of Dorylaimida and are limited to the families of Longidoridae and Trichodoridae [11]. Few species of these groups have been shown to transmit viruses in raspberry (Table 2), but several include virus vectors in blackberry and is thus likely to harbor potential vectors in raspberry as well.

### 4.1. Aphids

Aphids are well-known vectors of various plant viruses. There are approximately 300 species of aphids identified as vectors. They are such efficient virus vectors because they can transmit virus in all modes of transmission, including, non-persistent (stylet-borne), semi-persistent (foregut-borne), persistent circulative, and persistent propagative [69]. However, most species of aphids transmit virus through the stylet-borne non-persistent mechanism [3].

The European large raspberry aphid, *Amphorophora* (*Am.*) *idaei* (Börner), is the most economically important aphid pest in commercially grown red raspberry in Northern Europe and the United Kingdom (U.K.) [70]. This is because it is effective in transmitting the viruses in the raspberry mosaic disease (RMD) complex: black raspberry necrosis virus (BRNV), raspberry leaf mottle virus (RLMV) [raspberry leaf spot virus (RLSV)], and rubus yellow net virus (RYNV) [61,64]. RLMV and RLSV were previously considered as two separate viruses because of their differences in symptoms but due to their genetic similarities, they are now considered as isolates of the same virus [71]. The transmission of these viruses is likely to be semi-persistent [70]. Furthermore, the findings of McMenemy, et al. [72] suggested that BRNV and RLMV can make infected raspberry plants more attractive to aphids and prolong the aphid development time on infected plants to increase chances of virus acquisition, but the attraction is short-lived. Another aphid, *Amphorophora* (*Am.*) *rubi* (Kaltenbach), which is widely distributed in Europe and New Zealand [61], is also capable of transmitting RLMV and their isolate (RLSV) [31]. It is often found on blackberry and rarely on raspberry [61,73], therefore, it cannot be excluded as a raspberry virus vector. *Amphorophora idaei* and *Am. rubi* are often difficult to distinguish morphologically [73], and the host plant is often used to ease the identification process [74]. Therefore, it is necessary to develop a reliable barcoding method for better determination of these species.

The American species of large raspberry aphid, *Amphorophora* (*Am.*) *agathonica* Hottes, is closely related to *Am. idaei*, and the prevalent aphid vector for raspberry viruses in North America. It is capable of transmitting BRNV, RLMV, RYNV, and raspberry latent virus (RpLV) [65]. Unlike the former three viruses, RpLV is transmitted in a persistent propagative manner, but the efficiency of transmission is low. In addition, a co-infection of RLMV and RpLV is commonly found in the field [39]. This co-infection has neither synergistic nor antagonistic interaction between the two viruses, and *Am. agathonica* does not show any positive preference towards co-infected plants [75]. However, the raspberry plant exhibits a higher yield loss when it is co-infected with RLMV and RpLV [76]. Therefore, effective control of *Am. agathonica* is vital in reducing raspberry yield loss. Both *Am. idaei* and *Am. agathonica* have evolved new resistance-breaking biotypes under the selection pressure exerted by aphid-resistant raspberry cultivars. This poses a threat to commercial raspberry production [77]. However, the mechanism of aphids’ resistance against the raspberry aphid-resistance gene is still poorly understood, and thus, more in-depth studies are encouraged to overcome the threat of new resistant biotypes and maintain the effectiveness of raspberry aphid management [65].

Another economically important raspberry virus aphid-vector is the small raspberry aphid, *Aphis* (*Ap.*) *idaei* van der Goot, which occurs in the U.K., Europe, Canada, and New Zealand [78]. It is known only to transmit raspberry vein chlorosis virus (RVCV) [46]. RVCV is a persistent propagative virus, which makes the vector potentially infectious throughout its life [30]. Densely packed colonies of *Ap. idaei* are typically found on the tips of young canes and on the leaf petioles during spring, whereas scattered, small individuals are commonly found on the lower surface of the leaves throughout summer. The densely packed *Ap. idaei* colonies are more efficient in virus transmission than dispersed individuals [79]. Two non-EU raspberry viruses that are transmitted by an arthropod vector are RpLV and raspberry leaf curl virus (RpLCV) [29,36,59]. The small raspberry aphid, *Aphis* (*Ap.*) *rubicola* Oestlund, which is reported only in North America, is believed to be the only known vector of RpLCV [30,36]. Thus far, the RpLCV is only present in North America [80]. *Aphis rubicola* is reported to be an inefficient vector for RpLCV, because even under optimum conditions with an increasing population, the number of infected plants remains low [81]. However, there is a lack of recent studies on the status of both *Ap. idaei* and *Ap. rubicola* as pests and vectors in raspberry.

The potato aphid, *Macrosiphum euphorbiae* (Thomas), is occasionally seen infesting raspberry plants, but it causes minor damage to the crop [61]. It is usually found during spring and early summer on the leaf petioles and fruiting laterals of raspberry, but will eventually migrate to other crops when the population grows larger later in the summer [74]. On the other hand, another study reported that individuals of *M. euphorbiae* were found regularly along with *Am. idaei* on the raspberry grown under plastic tunnel until the end of first harvest. However, no population increase or significant damage was observed during the study when parasitoids *Aphidius* sp. Nees and *Praon volucre* (Haliday) (Hymenoptera: Braconidae) were released to suppress the aphid populations [82]. Apart from *M. euphorbiae*, the blackberry-cereal aphid, *Macrosiphum fragariae* (Walker), which is also known as *Sitobion fragariae*, may be present on raspberry plants during autumn (around October). They colonize and lay eggs, but the eggs laid on raspberry are unable to mature in spring and thus, infestation usually fails to develop [61,74]. However, these two species of aphids and the ornate aphid, *Myzus ornatus* Laing, were found capable of transmitting several viruses of the RMD complex in Europe and North America [66]. Unfortunately, there are no recent studies on these species on raspberry and their role as vectors of raspberry viruses. The green peach aphid, *Myzus persicae* (Sulzer), which was previously unknown as a pest of raspberry, was suspected as the vector of sowbane mosaic virus—rubus strain (SoMV-R). However, this has not yet been proven [30].

### 4.2. Whiteflies

Whiteflies are also important plant virus vectors, transmitting many plant viruses of economic importance. They are usually present in regions with warm climate and in greenhouses [83]. Only a few of the 1500 species are capable in transmitting viruses, though, the two most prominent being the tobacco whitefly, *Bemisia tabaci* (Gennadius) complex, and the greenhouse whitefly *Trialeurodes vaporariorum* (Westwood). Other known virus vectors are castor bean whitefly, *Trialeurodes ricini* (Misra), banded-winged whitefly, *Trialeurodes abutiloneus* (Haldeman), *Bemisia afer* (Priesner & Hosny)*,* and spiraling whitefly, *Aleurodicus dispersus* Russell [84]. Whiteflies are able to transmit viruses in three modes: non-persistent, semi-persistent, and persistent circulative [69,84,85].

Although whiteflies do feed on raspberry, to date, there is no published evidence of whiteflies acting as vector of any raspberry virus. However, this does not exclude them from being a potential raspberry virus vector. This is because whiteflies vector two significant blackberry viruses, namely blackberry yellow vein-associated virus (BYVaV) and beet pseudo-yellows virus (BPYV) [30,36]. Both of these viruses are from genus *Crinivirus* (*Closteroviridae*) and thus transmitted in a semi-persistent manner [86,87]. Both *T. vaporariorum* and *T. abutiloneus* can be involved in transmitting BYVaV [88], but only the greenhouse whitefly has shown to be involved in the transmission of BPYV [89]. The BYVaV is so far common in the United States and infecting only blackberry plants [86,88]. Unlike the BYVaV, the BPYV is present in several continents across the globe, namely, Asia, Europe, North America, and Oceania [90]. Whiteflies are usually present when raspberry is planted in plastic tunnels [24]. For instance, in Michigan, *B. tabaci* and *T. abutiloneus* were found infesting raspberry, which was planted in a plastic tunnel [26]. Also, *T. vaporariorum* is an increasingly important pest for both indoor and outdoor raspberry crops [91], and black raspberry has been reported as the host of *A. dispersus* in coastal Kenya [92]. There is a definite risk that these whitefly species may transmit any virus in a non-persistent manner when moving between infected and healthy raspberry plants. Based on the CABI [80] database, *T. abutiloneus* is present in North America only, *A. dispersus* is widely distributed in Asia, Oceania, Africa, and America, and present in Spain and Portugal in Europe. *Trialeurodes vaporariorum* and *B. tabaci* are present in almost all countries across the globe.

### 4.3. Leafhoppers

Approximately 50 species of leafhoppers from 25 genera in the family of Cicadellidae have been discovered as vectors of different plant viruses. They transmit viruses in either semi-persistent, persistent circulative, or persistent propagative manner [69]. Generally, most viruses transmitted by leafhoppers are infecting cereal crops, such as maize and rice [4]. In blackberry, leafhoppers are suspected vectors of blackberry virus S (BLVS) because they are known to transmit most marafiviruses. However, this is not determined yet [30]. The most commonly found leafhopper in raspberry seems to be the rose leafhopper, *Edwardsiana rosae* (Linnaeus). It is known to only cause light damage due to feeding on the plants but not as a virus vector [61]. Likewise, several species of *Empoasca*, such as *E. flavescens* Fabricius [93], *E. decedens* Paoli [94], and *E. fabae* (Harris) [39], can infest raspberry, but none of them have been reported as virus vectors. The same applies to *Typhlocyba pomeria* McAtee, which was found in raspberry cultivated under plastic tunnel in North America [26].

Leafhoppers are, however, associated with a phytoplasma disease in raspberry, called rubus stunt. It affects multiple *Rubus* spp., including blackberry, black raspberry, loganberry, and dewberry. The abovementioned leafhoppers found on raspberry are not known as vectors of rubus stunt phytoplasma. Thus far, the only known vector associated with this disease is the rubus leafhopper, *Macropsis fuscula* (Zetterstedt) [95]. However, even with a low population of *M. fuscula* on raspberry and blackberry, the rubus stunt disease can still be wide spread [96]. Besides the phytoplasma disease, *M. fuscula* along with the potato leafhopper, *E. fabae*, was suspected as the vector for RpLV, but this has later been proven false [39]. Despite being reported as a common leafhopper on raspberry in the Netherlands [74], it has been noted from Germany and Italy that *M. fuscula* and other *Macropsis* spp. are quite rare on raspberry and blackberry [96,97].

### 4.4. Thrips

Thrips are well-known as vectors for tospoviruses. The tospoviruses are transmitted in a persistent propagative manner, where the viruses must be acquired during first and early second larvae instars, otherwise it is unable to be transmitted [98,99]. However, thrips may have a preference for feeding and reproducing on tospovirus-infected plants, and thus increasing the chances of larvae acquiring the virus [100]. More in-depth studies are needed to improve understanding of these virus-vector interactions. Besides tospoviruses, thrips are known as vectors of plant viruses in the genera of *Ilarvirus, Carmovirus*, *Sobemovirus,* and *Machlomovirus*. The first three of these are pollen-borne, and the thrips physically carry the infected pollen to another plant, while infection happens via feeding wounds [5,101]. To date, only five thrips genera are known to transmit viruses, namely *Thrips*, *Frankliniella*, *Scirtothrips*, *Microcephalothrips,* and *Ceratothripoides* [101]. More thrips vectors may yet be discovered.

At least five species of thrips have been reported in raspberry (Table 3). *Frankliniella occidentalis* (Pergande) and *Thrips tabaci* Lindeman are widely distributed globally, whereas the three others have a more limited distribution [80]. *Thrips frici* (Uzel) is present in southern Europe, New Zealand, southern Australia, and some countries in America [102]. *Thrips imaginis* Bagnall is reported only in Oceania [103], and *Thrips fuscipennis* Haliday is present in Europe and North America [80,104]. Among these five species, *F. occidentalis* and *T. tabaci* are well known as vectors of plant viruses. In contrast, *T. frici* is unable to transmit tomato spotted wilt virus (TSWV) and impatiens necrotic spot virus (INSV), excluding it as vector for these tospoviruses [105]. *Thrips imaginis* and *T. fuscipennis* are also not proven as vectors of tospovirses [106,107,108]. However, *T. imaginis* is involved in transmitting pollen-borne viruses such as prunus necrotic ringspot virus (PNRSV) in stone fruit [109]. There is yet no report of tospovirus infecting red raspberry, but INSV has been found infecting blackberry. *Frankliniella occidentalis* shows high efficiency in transmitting the INSV under experimental conditions, but more studies are needed to obtain more evidence on the role of *F. occidentalis* in the spread of this virus [30]. Tobacco streak virus *Rubus* strain (TSV-R) is another virus that infects blackberry, and also black raspberry. TSV-R (*Ilarvirus*) is recognized as a pollen or seed-borne virus, but thrips, such as *F. occidentalis* and *T. tabaci*, are known vectors for some other strains of TSV [110]. It is suspected that these thrips may be involved in transmission of TSV-R by transporting the virus-carrying pollen to another plant, aiding the virus infestation. No conclusion can be made until more evidence is found. Likewise, the feeding of these thrips on red raspberry may also be involved in direct or indirect transmission of viruses, which have yet to be discovered. In addition, there are probably more thrips species feeding on raspberry than the ones listed in Table 3, for example, *Thrips major* Uzel, *Thrips flavus* Schrank, and *Frankliniella intonsa* (Trybom) have been found on other cultivated *Rubus* spp. [111,112]. Therefore, more effort in the identification of thrips on raspberry is encouraged.

### 4.5. Mites

Mite virus vectors typically belong to the families of Eriophyidae, Tetranychidae, and Tenuipalpidae. These mites are known to associate with viruses in the genera *Caulimovirus*, *Crinivirus*, *Luteovirus*, *Geminivirdae*, *Reovirus*, *Tospovirus,* and *Tenuivirus*, but some tetranychid and eriophyid mites are also known as vectors for emaraviruses, rymoviruses, allexiviruses, trichoviruses, poacevirus, Timbo virus (TIMV), and a nepovirus [60]. Mites from all the above-mentioned families have been recorded infesting raspberry (Table 2 and Table 3). The two-spotted spider mite, *Tetranychus urticae* Koch, the European red mite, *Panonychus ulmi* (Koch) (Acari: Tetranychidae) and the raspberry leaf and bud mite, *Phyllocoptes gracilis* (Nalepa) (Acari: Eriophyidae) are the most common mite species infesting raspberry [61,113]. *Tetranychus urticae* was the only mite found in a recent study carried out in raspberry grown under plastic tunnels in North America [26], even though the other two abovementioned species are also present in the continent [61]. However, in another recent survey, in Serbia, five spider mite species were found infesting raspberry: *Amphitetranychus viennensis* (Zacher), *Eotetranychus rubiphilus* (Reck), *Neotetranychus rubi* Trägårdh, *Tetranychus turkestani* Ugarov and Nikolski, and *T. urticae* [114]. The varying species of mites found in different locations is most probably due to the species distribution. For instance, *A. viennensis* is only found in Asia and Europe, while *T. urticae* is more widely distributed across different continents [80]. Local surveys of mites are important to obtain an overview of the species present.

Although several species of mites are feeding on raspberry, most of them are only known to cause physical damage. To date, *P. gracilis*, is the only mite associated with a raspberry virus, specifically to raspberry leaf blotch virus (RLBV) (genus *Emaravirus*) [41]. The mite was first suspected to associate with this virus based on the feeding symptoms, such as irregular yellow blotches on leaves, curling of leaves, and distortion of leaf margins, which commonly appear as symptoms of viral infection [115]. These symptoms were earlier known as raspberry leaf blotch disorder (RLBD). RLBV was named in the studies by McGavin, Mitchell, Cock, Wright, and MacFarlane [67], where they also found strong evidence of *P. gracilis* as the vector of this virus. Similar evidence was found by Dong, Lemmetty, Latvala, Samuilova, and Valkonen [41], where *P. gracilis* was present in all the plants exhibiting RLBD symptoms and RLBV RNA was detected in the mites. These mites also demonstrated the ability to transmit the virus suggesting it to be a natural vector, but further studies on the transmission mechanism are recommended. On top of virus transmission, the physical damage of high *P. gracilis* populations also lead to high yield loss. For instance, they may feed on developing raspberries causing premature ripening of some drupelets and irregular shaped fruits which are hardly marketable [116]. While no other mites on raspberry are reported as a virus vector currently, they do possess the capability of transmitting viruses. Therefore, more studies of the mite involvement in viral transmission should be carried out.

### 4.6. Plant-Parasitic Nematodes

In general, the process of virus transmission by nematodes is divided into six phases, which begin with (1) ingestion of virus particles from an infected plant, (2) acquisition, (3) adsorption, (4) retention, (5) release of virus from retention site in the nematode, and lastly (6) transfer and establishment, where the virus particles are transferred to healthy plant cells and replication of these viruses occur which lead to a successful infection [5]. The plant-parasitic nematodes from the order of Dorylaimida and Triplonchida are proven to transmit plant viruses, mostly tobraviruses and nepoviruses [11]. Thus far, only 14 out of approximately 75 species in the genera of *Trichodorus* and *Paratrichodorus* from the order of Triplochida, and a few out of 350 described species in the genera of *Xiphinema*, *Longidorus,* and *Paralongidorus* from the order of Dorylaimida have been proven as plant virus vectors [9]. Plant-parasitic nematodes from the order of Triplochida have not been found on raspberry plants. However, from the order of Dorylaimida, eight species in the genera of *Longidorus* (needle nematodes) and *Xiphinema* (dagger nematodes), have been reported as pests of raspberry. These nematodes are ectoparasitic, feeding on the outside of the root. Six species of longidorid-nematodes, namely *Longidorus attenuatus* Hooper, *L. elongatus* (de Man) Thorne and Swanger, *Xiphinema americanum* Cobb, *X. diversicaudatum* (Micoletzky) Thorne, *X. pachtaicum* (Tulaganov) Kirjanova, and *X. vuittenezi* Luc, Lima, Weischer and Flegg have been reported from more than one continent, all of them being present in Europe [44,80,117]. *Longidorus macrosoma* Hooper has only been reported from Europe and *X. bakeri* (Williams) from America only [80,118].

The plant viruses associated with longidorid-nematodes are mainly belonging to genus *Nepovirus*. Besides this, one species from the genus *Cheravirus* (cherry rasp leaf virus [CRLV]) and another unassigned species in the family of *Secoviridae* (strawberry latent ringspot virus [SLRSV]) have also been reported [9,33]. The SLRSV was recently proposed to a new genus “*Stralarivirus*” within the family of *Secoviridae* [51]. Raspberry is infected by eight nematode-transmitted viruses and all these viruses are from the *Secoviridae* family (Table 1). Among these viruses, six of them are from genus *Nepovirus*, namely, arabis mosaic virus (ArMV), cherry leafroll virus (CLRV), raspberry ringspot virus (RpRSV), tobacco ringspot virus (TRSV), tomato black ring virus (TBRV), and tomato ringspot virus (ToRSV). The remaining two other viruses are CRLV and SLRSV [30,33,54]. All the longidorid-nematodes found on raspberry, except *X. pachtaicum*, are associated with at least one of the abovementioned viruses. *Xiphinema diversicaudatum* is known as the vector of both ArMV and SLRSV [119,120]. The mixed infection of ArMV and SLRSV will lead to raspberry yellow dwarf disease [30]. Besides *X. diversicaudatum*, ArMV may also be transmitted by *X. bakeri* [68], but the evidence is insufficient [121]. *Xiphinema americanum* is associated with several viruses infecting *Rubus*. It is known as a vector of ToRSV, CLRV, and CRLV in raspberry [36,59,122] and vector of tobacco ringspot virus (TRSV) in blackberry [123,124]. Among the three raspberry viruses transmitted by *X. americanum*, ToRSV and CLRV are economically important in raspberry cultivation, while the economic impact of CRLV is more uncertain [30,125]. ToRSV infected plants display weakened vigor, decrease in fruit yield and quality, and in the long run, the plant will be stunted and eventually die [59,126]. CLRV infected plants will also be stunted with distorted leaf development and in addition, the leaves of fruiting canes may exhibit chlorotic mottle and ringspot [31]. This will eventually lead to a significant decline in fruit productivity. *Xiphinema vuittenezi* is the only species found transmitting TRSV in raspberry in Slovakia [54]. Lastly, the TBRV and RpRSV were transmitted by three *Longidorus* sp. nematode, where both viruses are transmitted by *L. elongatus*, whereas *L. attenuatus* is only known to transmit TBRV and *L. macrosoma* only transmit RpRSV [44,45,120]. Since both TBRV and RpRSV are vectored by *L. elongatus*, a mixed infection of these viruses is often observed and usually results in raspberry leaf curl disease [127]. Despite the number of nematode vectors reported, their role is still understudied due to their cryptic habitat and small size posing a challenge in detecting and verifying transmission. Undoubtedly, many nematode vectors of viruses are yet to be identified.

### 4.7. The Interaction between Virus Vectors and Other Herbivores

Besides the groups with known or potential virus vectors discussed above, a lot of other arthropods are feeding on raspberry, including beetles (Coleoptera), moths (Lepidoptera), flies (Diptera), and sawflies and gall wasps (Hymenoptera), in addition to several groups of true bugs (Hemiptera) (Table 3). All these could indirectly influence the spread of viruses through competitive interactions with virus vectors. In general, competitors could be expected to reduce the vector fitness (i.e., lower fecundity and higher mortality) and hence reducing the spread of viruses. In addition, this interaction could also decrease the feeding time, thus, reducing the chances of vectors acquiring or introducing plant viruses [128]. However, competitive interaction can also enhance the spread of viruses. To avoid competition, vectors may disperse to other individual plants to achieve higher fitness. A vector that has acquired virus from an infected plant may then land on a healthy plant, thus leading to new infection [129]. For instance, interactions between the pea leaf weevil beetle, *Sitona lineatus* (Linnaeus), and the pea aphid, *Acyrthosiphon pisum* (Harris), on pea plants have been found to promote the spread of pea enation mosaic virus (PEMV) [130]. This is mainly due to the presence of *S. lineatus* increasing the reproduction of *A. pisum* causing crowding and thus encouraging individuals of *A. pisum* to migrate to other pea plants, indirectly promoting the spread of PEMV. In addition, *S. lineatus* promotes the spread of PEMV by displacing *A. pisum* to the areas on individual plants more susceptible to virus infection [130]. In another study, *T. urticae*, a non-vector in this case, was found aiding the spread of tomato yellow leaf curl virus (TYLCV) by suppressing two flavonoids in tomato plants which deter *B. tabaci* (the vector of TYLCV). This encourages more *B. tabaci* to feed on tomato plants and increase the TYLCV transmission [131]. Similar scenarios may be happening between non-vector herbivores and virus vectors on raspberry. Other interactions are also possible. For example, the feeding of weevil larvae (*Otiorhynchus sulcatus*) on raspberry roots has been shown to boost the population of the large raspberry aphid, *Am. idaei* [132]. Plant-parasitic nematodes can also affect the fitness of shoot-feeding insect pests. For example, higher fecundity of green peach aphid, *Myzus persicae* (Sulzer) is observed in potato crops pre-infected with a plant-parasitic nematode, *Globodera pallida* (Stone) [133]. This will promote the spread of potato plant viruses, such as potato virus Y (PVY), where *M. persicae* is the most important vector [134]. This means that even below-ground herbivores may affect the virus spread above-ground. The role of prevalent non-vectors should not be neglected when developing viral pathogen management strategies.

**Table 3 viruses-14-00571-t003:** Potential vector and non-vector invertebrate herbivores feeding on raspberry plants (*Rubus idaeus*).

Herbivore Group	Family	Species	References
Aphids	Aphididae	*Acyrthosiphon malvae* (Mosley) *Amphorophora amurensis* (Mordvilko) *Amphorophora sensoriata* Mason *Aphis gossypii* Glover *Aphis ruborum* (Börner & Schilder) *Kaltenbachiella pallida* (Haliday) *Macrosiphum funestum* (Macchiati) *Matsumuraja hirakurensis* Sorin *Matsumuraja rubi* (Matsumura) *Matsumuraja rubifoliae* Takahashi *Matsumuraja taisetsusana* Miyazaki *Pemphigus rubiradicis* Theobald	[61,62,135]
Whiteflies	Aleyrodidae	*Aleurodicus dispersus* Russell *Aleyroides lonicerae* Walker *Bemisia tabaci* (Gennadius) *Trialeurodes abutiloneus* (Haldeman) *Trialeurodes vaporariorum* (Westwood)	[26,91,92,136]
Leafhoppers	Cicadellidae	*Edwardsiana rosae* (Linnaeus) *Edwardsiana sociabilis* (Ossiannilsson) *Empoasca* spp. Walsh *Evacanthus interruptus* (Linnaeus) *Macropsis fuscula* (Zetterstedt) *Platymetopius undatus* (De Geer) *Typhlocyba pomaria* McAtee *Ribautiana tenerrima* (Herrich-Schaffer)	[26,61,74,137,138,139]
Treehopper	Membracidae	*Centrotus cornutus* (Linnaeus)	[26,61,62,63,113,135,140]
Spittlebug	Aphrophoridae	*Philaenus spumarius* (Linnaeus)	[26,91,92]
Capsid bugs	Miridae	*Closterotomus fulvomaculatus* (De Geer) *Lopidea dakota* Knight *Lygocoris pabulinus* (Linnaeus) *Lygus lineolaris* (Palisot de Beauvois) *Lygus rugulipennis* Poppius *Plagiognathus arbustorum* (Fabricius)	[26,61,141,142]
Shield bugs	Pentatomidae	*Cuspicona simplex* Walker *Dolycoris baccarum* (Linnaeus) *Nezara viridula* (Linnaeus) *Palomena prasina* (Linnaeus) *Pentatoma rufipes* (Linnaeus) *Plautia affinis* Dallas	[61,143]
	Pyrrhocoridae	*Dindymus versicolor* (Herrich-Schaeffer)	[143]
	Coreidae	*Amblypelta nitida* Stål	[143]
Scale Insects	Coccidae	*Aulacaspis rosae* (Bouché) *Parthenolecanium corni* (Bouché)	[61,144]
Cicada	Cicadidae	*Cicadetta montana* (Scopoli)	[113]
Tree Crickets	Gryllidae	*Oecanthus nigricornis* (Walker) *Oecanthus pellucens* (Scopoli)	[63,113]
Thrips	Thripidae	*Frankliniella occidentalis* (Pergande) *Tenothrips frici* (Uzel) *Thrips imaginis* Bagnall *Thrips fuscipennis* Haliday *Thrips tabaci* Lindeman	[26,143,145,146]
Beetles	Attelabidae	*Neocoenorrhinus germanicus* (Herbst)	[61]
	Buprestidae	*Agrilus cuprescens* (Ménétriés) (syn.: *A. aurichalceus* Redtenbacher) *Agrilus ruficollis* (Fabricius) *Coraebus rubi* (Linnaeus)	[61,63,113,140]
	Byturidae	*Byturus rubi* Barber *Byturus tomentosus* (De Geer) *Byturus unicolor* Say	[26,61,63,113,147]
	Cantharidae	*Cantharis obscura* Linnaeus	[61]
	Cerambycidae	*Oberea bimaculata* (Olivier)	[63]
	Chrysomelidae	*Batophila aerate* (Marsham) *Batophila rubi* (Paykull) *Galerucella sagittariae* (Gyllenhal)	[61]
	Curculionidae	*Anthonomus rubi* (Herbst) *Barypeithes araneiformis* (Schrank) *Barypeithes pellucidus* (Boheman) *Mitoplinthus caliginosus* (syn.: *Plinthus caliginosus*) (Fabricius) *Otiorhynchus armadillo* (Rossi) *Otiorhynchus clavipes* (Bonsdorff) *Otiorhynchus globus* Boheman *Otiorhynchus ovatus* (Linnaeus) *Otiorhynchus rugosostriatus* (Goeze) *Otiorhynchus singularis* (Linnaeus) *Otiorhynchus sulcatus* (Fabricius) *Peritelus noxius* Boheman *Sciaphilus asperatus* (Bonsdorff)	[61,63,113,148,149]
	Elateridae	*Agriotes lineatus* (Linnaeus) *Agriotes obscurus* (Linnaeus)	[61]
	Scarabaeidae	*Cetonia aurata* (Linnaeus) *Cotinis nitida* (Linnaeus) *Macrodactylus subspinosus* (Fabricius) *Melolontha melolontha* (Linnaeus) *Popillia japonica* Newman *Tropinota hirta* (Poda) *Amphimallon solstitialis* (Linnaeus)	[61,63,113,140]
	Tenebrionidae	*Lagria hirta* (Linnaeus)	[61]
Moths	Cossidae	*Zeuzera pyrina* (Linnaeus)	[113]
	Erebidae	*Arctia caja* (Linnaeus) *Euproctis similis* (Fuessly) *Lymantria dispar* (Linnaeus) *Orgyia antiqua* (Linnaeus) *Sphrageidus similis* (syn.: *Euproctis similis*) (Fuessly) *Spilosoma lutea* (Hufnagel)	[61,113,150]
	Geometridae	*Dysstroma truncata* (syn.: *Chloroclysta truncata*) (Hufnagel) *Operophtera bruceata* (Hulst) *Operophtera brumata* (Linnaeus) *Operophtera occidentalis* (Hulst) ^1^	[61,151,152]
	Hepialidae	*Hepialus humuli* (Linnaeus) *Hepialus lupulinus* (Linnaeus)	[61]
	Incurvariidae	*Lampronia rubiella* (syn.: *Incurvaria rubiella*) (Bjerkander)	[61,63,113]
	Lasiocampidae	*Macrothylacia rubi* (Linnaeus) *Malacosoma neustria* (Linnaeus)	[61,153]
	Nepticulidae	*Stigmella aurella* (Fabricius) *Stigmella fragariella* (Heinemann)	[61,150]
	Noctuidae	*Acronicta psi* (Linnaeus) *Ceramica pisi* (Linnaeus) *Graphiphora augur* (Fabricius) *Hydraecia micacea* (Esper) *Lacanobia oleracea* (Linnaeus) *Melanchra persicariae* (Linnaeus) *Naenia typica* (Linnaeus) *Orthosia gothica* (Linnaeus) *Orthosia gracilis* (Denis & Schiffermüller) *Orthosia incerta* (Hufnagel) *Papaipema nebris* (Guenée) *Peridroma saucia* (Hübner) *Xestia c-nigrum* (Linnaeus)	[61,63,113,150]
	Notodontidae	*Phalera bucephala* (Linnaeus)	[61]
	Oecophoridae	*Carcina quercana* (Fabricius)	[61]
	Saturniidae	*Saturnia pavonia* (Linnaeus)	[61,113]
	Schreckensteiniidae	*Schreckensteinia festaliella* (Hübner)	[61]
	Sesiidae	*Pennisetia hylaeiformis* (Laspeyres) *Pennisetia bohemica* Králíček & Povolný *Pennisetia marginata* (Harris) *Synanthedon bibionipennis* (Boisduval)	[61,63,113,154,155]
	Thyatiridae	*Thyatira batis* (Linnaeus)	[61]
	Tischeriidae	*Tischeria marginea* (syn.: *Coptotriche marginea*) Haworth	[61]
	Tortricidae	*Acleris laterana* (Fabricius) *Acleris variegana* (Denis & Schiffermüller) *Adoxophyes orana* (Fischer von Röslerstamm) *Archips podana* (Scopoli) *Archips rosana* (Linnaeus) *Argyrotaenia citrana* (Fernald) *Cacoecimorpha pronubana* (Hübner) *Celypha lacunana* (Denis & Schiffermüller) *Choristoneura rosaceana* (Harris) *Clepsis spectrana* (Treitschke) *Cnephasia asseclana* (Denis & Schiffermüller) *Cnephasia longana* (Haworth) *Ditula angustiorana* (Haworth) *Epiphyas postvittana* (Walker) *Lozotaenia forsterana* (Fabricius) *Pandemis cerasana* (Hübner) *Pandemis heparana* (Denis & Schiffermüller) *Ptycholoma lecheana* (Linnaeus) *Notocelia uddmanniana* (Linnaeus) *Spilonota ocellana* (Denis & Schiffermüller)	[61,63,113,140,150,156,157,158]
Flies	Agromyzidae	*Agromyza potentillae* (Kaltenbach)	[61]
	Anthomyiidae	*Pegomya rubivora* (Coquillett)	[61,63,113]
	Cecidomyiidae	*Dasineura plicatrix* (Loew) *Resseliella theobaldi* (syn.: *Thomasiniana theobaldi*) (Barnes) *Lasioptera rubi* (Schrank)	[61,63,113,140,156,159,160]
	Drosophilidae	*Drosophila suzukii* (Matsumura)	[26,161]
	Tipulidae	*Nephrotoma appendiculata* (Pierre)	[61]
Sawflies	Tenthredinidae	*Allantus cinctus* (Linnaeus) *Cladius difformis* (Panzer) *Empria tridens* (Konow) *Metallus pumilus* (Klug) *Monophadnoides geniculatus* (Hartig) *Priophorus morio* (Lepeletier)	[61,63]
	Cephidae	*Hartigia cressoni* (Kirby)	[162]
Gall wasp	Cynipidae	*Diastrophus rubi* (Bouché)	[61]
Mites	Tetranychidae	*Amphitetranychus viennensis* (Zacher) *Eotetranychus carpini borealis* (Ewing) *Eotetranychus frosti* (McGregor) *Eotetranychus rubiphilus* (Reck) *Neotetranychus rubi* Trägårdh *Neotetranychus rubicola* Bagdasarian *Panonychus ulmi* (Koch) *Tetranychus mcdanielli* McGregor *Tetranychus schoenei* McGregor *Tetranychus turkestani* Ugarov & Nikolski *Tetranychus urticae* Koch	[26,61,63,113,114,156,163,164,165,166]
	Tenuipalpidae	*Cenopalpus spinosus* (Donnadieu) *Pentamerismus erythreus* (Ewing)	[164,167]
	Eriophyidae	*Acalitus essigi* (Hassan) *Acalitus orthomera* (Keifer) *Aceria silvicola* (Canestrini) *Epitrimerus gibbosus* (Nalepa)	[113,168]
Nematodes (Order: Tylenchida)	Anguinidae	*Ditylenchus dipsaci* (Kühn)	[117]
	Belonolaimidae	*Tylenchorhynchus elegans* Siddiqi *Tylenchorhynchus cylindricus* Cobb *Tylenchorhynchus claytoni* Steiner	[117]
	Criconematidae	*Xenocriconemella macrodora* (Taylor)	[117]
	Heteroderidae	*Meloidogyne arenaria* (Neal) *Meloidogyne hapla* Chitwood *Meloidogyne incognita* (Kofoid & White) *Meloidogyne javanica* (Treub)	[117,169]
	Hoplolaimidae	*Helicotylenchus digonicus* Perry *Helicotylenchus dihystera* (Cobb)	[117]
	Pratylenchidae	*Pratylenchus crenatus* Loof *Pratylenchus penetrans* (Cobb) *Pratylenchus scribneri* Steiner *Pratylenchus thornei* Sher & Allen *Pratylenchus vovlasi* sp. Nov. *Pratylenchus vulnus* Allen & Jensen	[63,117,118,169,170]
Nematode (Order: Dorylaimida)	Longidoridae	*Xiphinema pachtaicum* (Tulaganov) Kirjanova	[117]

^1^*Operophtera occidentalis* (Hulst) is treated as a subspecies of *O. bruceata* (Hulst) by Troubridge and Fitzpatrick [171].

## 5. Pest Management for Better Control of Raspberry Viruses

Aphids, mites, and nematodes are the only groups proven to be involved in raspberry virus transmission. In the effort to suppress the spread of raspberry viruses, the management of these vectors plays a vital role. Their small body size and cryptic lifestyle means that low abundances may be difficult to spot before virus symptoms appear. Also, to prevent virus transmission is more demanding than managing the vectors as ordinary pests, and there is a risk that failed control efforts can increase the virus transmission instead of reducing it [172]. Using plant material free of both viruses and vectors, and having detailed knowledge of the agroecosystem, is necessary for successful vector management.

### 5.1. Aphids

Aphid management in raspberry can be categorized into four: (1) breeding of aphid resistant cultivars, (2) chemical control, (3) biological control, and (4) other methods. Great effort has been made to overcome the threat of *Am. idaei* and *Am. agathonica* as vectors by developing aphid resistant raspberry cultivars [173]. However, subjected to this selection pressure, *Am. idaei* and *Am. agathonica* populations have evolved into biotypes that can overcome such plant resistance (better known as resistance-breaking biotypes) [77,174]. Cultivars resistant to *Ap. idaei* or *Ap. rubicola* have not been developed. Chemical control of raspberry aphids with insecticides, such as organophosphates, carbamate, neonicotinoids, pyrethroids, and butenolides, have been recommended to prevent spread of viruses in raspberry [74,175]. However, due to health and environmental hazards, many of these conventional insecticides are heavily restricted in Europe [176,177,178], and the EU aims to halve the pesticide use by 2030. There are still aphidicides available, but they often lack the knockdown or systemic effect necessary to eliminate aphids quickly [70], or are not allowed during flowering in order to protect pollinators [74,176]. Fewer available pesticides means a higher risk of breeding insecticide-resistant aphid populations [179]. To sum up, the role of chemical control in aphid management is likely to decrease. This may not affect the virus management as much as expected, as it has long been known that effective chemical control of the large raspberry aphid does not necessarily lead to a significant reduction in the viruses it transmits [180].

In terms of biological control, aphids in raspberry have many natural enemies, such as parasitoids, ladybeetles, lacewings, and entomopathogenic microbes [70,82,181]. A combination strategy using aphid-resistant cultivars and commercially available aphid parasitoids (*Aphidius ervi* Haliday) seems promising, although aphid-parasitoid interactions are affected by the resistance [182]. Commercially available microbials, like *Beauveria bassiana* (Bals.-Criv.) Vuill, *Burkholderia* spp. and *Chromobacterium subtsugae* Martin et al., are recommended against *Am. agathonica* in the US [175]. In the UK, *Lecanicillium longisporum* (Petch) Zare and W.Gams, *Isaria fumosorosea* (Wize) Brown and Smith, and *Metarhizium brunneum* Petch have been found effective in managing *Am. idaei* populations in potted raspberry grown under glasshouse conditions [183]. Other methods for aphid control include various types of nets [184] and traps [185], physically acting insecticides [186], semiochemical repellents [187], and barrier plants [188]. These are little used in raspberry, and not all are well suited, but exclusion nets and repellents seem of particular interest if the main goal is to avoid virus transmission. There is a need to know more about the combined effect of new and old control measures in raspberry pest management, both on virus transmission and pest abundances in general. Control of weeds and fungi may also significantly impact vector management, for example by fungicides reducing the effect of entomopathogenic fungi.

### 5.2. Mites

*Phyllocoptes gracilis* is the only mite associated with a raspberry virus. In the 1990s, this mite was effectively managed using broad spectrum insecticides, such as systemic organophosphates and endosulfan or acaricides, such as bromopropylate. Most of these pesticides have later been banned due to environmental and health impact, but some newer acaricides, like spirodiclofen and fenpyroximate also have effect [189]. In addition, it is possible to target the overwintering females with late autumn sprays with sulfur or vegetable oil [190,191]. Biological control using predatory mites (Acari: Phytoseiidae), such as *Amblyseius andersoni* (Chant), *Typhlodromus pyri* Scheuten, and *Phytoseius macropilis* (Banks), can provide a good suppression of *P. gracilis* population [74,192]. Unfortunately, these predatory mites are unable to provide sufficient control of *P. gracilis* population under all raspberry growth conditions. Some of these phytoseiid species are classified as food-generalists, where they also feed on pollen, fungi, and plant exudates [193]. For example, *T. pyri* has been found feeding on apple leaves and fruits as well as apple powdery mildew, even in the presence of pollen and prey [194,195]. This could potentially facilitate the transmission of plant viruses either directly or via virus-infected fungi [196]. On the other hand, having a flexible diet allows these mites to persist in the absence of a specific prey and hence, possibly providing sustainable control once this prey occurs [193,194]. Further studies on plant feeding behavior of predatory mites should be carried out to better understand their role as predators of *P. gracilis*. Entomopathogenic fungi, such as *B. bassiana* and *Metarhizium anisopliae* (Metschn.) Sorokīn, have been assessed as potential biological control agents to provide additional control of this mite. The preliminary results of this assessment seem promising, but more studies have to be carried out to confirm the efficacy [197]. Effective management of *P. gracilis* populations will result in less RLBV in raspberry plantations.

### 5.3. Nematodes

In general, the control measures of nematodes can be divided into: (1) chemical approaches, (2) cultural approaches, (3) host plant resistance, and (4) biological control. In chemical approaches, fumigants, such as dichloropropane-dichloropropene, methylbromide, and dazomet, and non-fumigants, such as oxime-carbamate, organophosphate, and methylcarbamates, were recommended. This has a negative environment and health impact, therefore, many of these pesticides were banned or are in the process of being phased out in the European Union (e.g., Regulations (EC) 2037/2000 and (EC) 1107/2009) [198]. Crop rotation and flooding of field if permitted can be used as cultural control methods. Another option is to cultivate nematode-resistant or related virus resistant host plant cultivars to overcome nematode damages. However, this may lead to the development of resistance-breaking nematodes or viruses [12,199]. In the EU, commercially produced plant extract, such as garlic extract, is also recommended and proven effective against plant-parasitic nematodes including *Longidorus* spp. and *Xiphinema* spp., which are vectors of raspberry viruses [200]. The biological control agents against plant-parasitic nematodes comprise nematophagous fungi, nematophagous bacteria, nematophagous mites, plant growth-promoting rhizobacteria, arbuscular mycorrhizal fungi, and predatory nematodes [201]. *Mononchoides fortidens* (Schuurmans-Stekhoven), *Mononchoides longicaudatus* Khera, and *Mononchoides gaugleri* Siddiqi, Bilgrami, and Tabassum are examples of predatory nematodes that prey on several plant-parasitic nematodes including *Longidorus* sp. and *X. americanum* [202]. Edible mushroom, *Pleurotus* spp., is found to produce toxin effective against several genera of plant-parasitic nematode and toxin from *Pleurotus citrinopileatus* Singer is more effective than other species [203]. These biological control agents can either provide direct or indirect protection to the plant roots. Unfortunately, there are very few studies on the interaction between plant-parasitic nematodes and their respective natural enemies in raspberry crop.

## 6. Conclusions

Twenty-two viruses have been reported to infect raspberry. Among the invertebrate herbivores found on raspberry, the aphids (*Am. idaei*, *Am. rubi*, *Am. agathonica*, *Ap. idaei*, *Ap. rubicola*, *M. euphorbiae*, *M. fragariae*, and *Myzus ornatus* and plant-parasitic nematodes (*L. attenuatus*, *L. elongatus*, *L. macrosoma*, *X. americanum*, *X. diversicaudatum*, *X. vuittenezi,* and *X. bakeri*) are the only proven vectors of raspberry viruses based on the current available literature. The eriophyid mite (*P. gracilis*) is suggested as the natural vector of raspberry leaf blotch virus (RLBV), but further studies on the transmission mechanism is required. Even though most of the invertebrate herbivores have not been reported as virus vectors, their potential involvement in the spread of raspberry viruses should not be overlooked. Also, the interaction between these pests and their respective natural enemies, such as predators, parasitoids, and entomopathogens, should be studied to develop integrated pest management strategies in raspberry plantation that can suppress the spread of viral pathogens. These strategies should also include the use of cultivars with a high degree of resistance to viruses and/or their vectors, and growing techniques that inhibit locally important vectors. As a conclusion, there are still multiple aspects in this topic which require further studies, so to have a better understanding on the complex interactions among the host plant, viral pathogens, invertebrate vectors, and non-vectors in the raspberry agroecosystem. Eventually, this will assist in development of better strategies to minimize losses caused by raspberry pests and pathogens.

## Figures and Tables

**Table 1 viruses-14-00571-t001:** The known raspberry (*Rubus idaeus*) viruses and their mode of transmission (MoT).

Virus Name	Family	Genus	MoT ^1^	References
Apple mosaic virus (ApMV)	*Bromoviridae*	*Ilarvirus*	P, S	[28]
Arabis mosaic virus (ArMV)	*Secoviridae*	*Nepovirus*	S, N	[29,30]
Blackberry virus Y (BVY)	*Potyviridae*	*Brambyvirus*	U	[30]
Black raspberry necrosis virus (BRNV)	*Secoviridae*	*Sadwavirus*	A	[31,32,33]
Cherry leaf roll virus (CLRV)	*Secoviridae*	*Nepovirus*	P, S, N	[31,34]
Cherry rasp leaf virus (CRLV)	*Secoviridae*	*Cheravirus*	N	[35,36]
Cucumber mosaic virus (CMV)	*Bromoviridae*	*Cucumovirus*	A, S	[29,37,38]
Raspberry bushy dwarf virus (RBDV)	unassigned	*Idaeovirus*	P, S	[29,31]
Raspberry latent virus (RpLV)	unassigned	unassigned	A	[39,40]
Raspberry leaf blotch virus (RLBV)	*Fimoviridae*	*Emaravirus*	M	[41,42]
Raspberry leaf curl virus (RpLCV)	unassigned	unassigned	A	[30,43]
Raspberry leaf mottle virus (RLMV)	*Closteroviridae*	*Closterovirus*	A	[31,40]
Raspberry ringspot virus (RpRSV)	*Secoviridae*	*Nepovirus*	P, S, N	[30,44,45]
Raspberry vein chlorosis virus (RVCV)	*Rhabdoviridae*	*Cytorhabdovirus*	A	[31,46]
Rubus yellow net virus (RYNV)	*Caulimoviridae*	*Badnavirus*	A	[47,48]
Sowbane mosaic virus (SoMV)	*Solemoviridae*	*Sobemovirus*	P, S	[30,49,50]
Strawberry latent ringspot virus (SLRSV)	*Secoviridae*	*Stralarivirus*	N	[30,51,52]
Strawberry necrotic shock virus (SNSV)	*Bromoviridae*	*Ilarvirus*	P, S	[36,53]
Tobacco ringspot virus (TRSV)	*Secoviridae*	*Nepovirus*	S, N	[54,55]
Tobacco streak virus (TSV)	*Bromoviridae*	*Ilarvirus*	P, S	[53,56,57]
Tomato black ring virus (TBRV)	*Secoviridae*	*Nepovirus*	P, S, N	[30,44,58]
Tomato ringspot virus (ToRSV)	*Secoviridae*	*Nepovirus*	P, S, N	[29,30,59]

^1^ Mode of Transmission, P: Pollen, S: Seed, N: Nematode, A: Aphid, M: Mites, U: Unknown.

**Table 2 viruses-14-00571-t002:** Known invertebrate vectors of raspberry (*Rubus idaeus*) viruses.

Vector Group	Family	Species	References
Aphids	Aphididae	*Amphorophora idaei* (Börner) *Amphorophora rubi* (Kaltenbach) *Amphorophora agathonica* Hottes *Aphis idaei* van der Goot *Aphis rubicola* Oestlund *Macrosiphum euphorbiae* (Thomas) *Macrosiphum fragariae* (syn. *Sitobion fragariae*) (Walker) *Myzus ornatus* Laing	[30,31,46,61,62,63,64,65,66]
Mites	Eriophyidae	*Phyllocoptes gracilis* (Nalepa) ^1^	[41,67]
Nematodes	Longidoridae	*Longidorus attenuatus* Hooper *Longidorus elongatus* (de Man) Thorne & Swanger *Longidorus macrosoma* Hooper *Xiphinema americanum* Cobb *Xiphinema bakeri* Williams *Xiphinema diversicaudatum* (Micoletzky) Thorne *Xiphinema vuittenezi* Luc, Lima, Weischer & Flegg	[44,54,59,63,68]

^1^ There is strong evidence of *P. gracilis* being the natural vector of raspberry leaf blotch virus (RLBV) [41,67], but further studies on the transmission mechanism are needed to confirm it.

## Data Availability

Not applicable.

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
