# Peer review of "Known and Potential Invertebrate Vectors of Raspberry Viruses"

_viruses, 2022, doi:10.3390/v14030571_

Round 1

Reviewer 1 Report

The manuscript entitled “Raspberry viruses and their known and potential invertebrate vectors” by Tan and colleagues is a thorough review on the well-characterized and some potential vectors of different raspberry viruses, including the transmission mechanisms, geographical distribution, vector interactions with other herbivores and their management. I find the review well written, interesting and enriched with useful information. In my opinion, the authors have adequately addressed all major points. In conclusion, I find the manuscript acceptable for publishing after a few corrections are done:

L13 (abstract) Are there any recent data on global production of Rubus spp.? If not, is it feasible to calculate the average production for a longer period of time?

L127 (Table 1) The word „plant“ might be omitted from the title.

L135 “are” should be added in the sentence: In addition, plant-parasitic nematodes which are known to transmit plant viruses belong to the order of Dorylaimida and ARE limited to the families of Longidoridae and Trichodoridae [11].

L146 “circulative”, like propagative, should not be in parentheses.

L154 This is the first mention of RLSV. Therefore, the virus full name should be added. Iti s not clear if the complex encompasses three or four viruses since RLSV is written in parentheses.

L160 The relation of RLMV and its isolate RLSV should be briefly clarified.

L186 colonies are instead of “colonies is”

L297 Thrips imaginis and T. fuscipennis are also not proven as vectors  instead of “is proven as a vector”

L374 I suggest: The plant viruses associated with longidorid-nematodes are mainly nepoviruses. Otherwise, it should be: „mainly belong to genus Nepovirus

L443 a space should be added in “O. bruceata

Unless otherwise recommended by the journal, virus higher taxon name should be written in italics and begins with a capital letter (ICTV). This differs from the convention in botany and zoology, in which taxon names above the level of genus are not italicized.

I would suggest adding one figure with photos of the most important raspberry virus vectors. It might be helpful for those virologists who are not very familiar with the topic.

Author Response

The manuscript entitled “Raspberry viruses and their known and potential invertebrate vectors” by Tan and colleagues is a thorough review on the well-characterized and some potential vectors of different raspberry viruses, including the transmission mechanisms, geographical distribution, vector interactions with other herbivores and their management. I find the review well written, interesting and enriched with useful information. In my opinion, the authors have adequately addressed all major points. In conclusion, I find the manuscript acceptable for publishing after a few corrections are done:

First, the authors would like to thank the reviewer for taking the effort to carefully review the manuscript. All the suggestions provided are valuable and we appreciate them very much. Therefore, we have accepted almost all the recommendations and revised them as best we could.

L13 (abstract) Are there any recent data on global production of Rubus spp.? If not, is it feasible to calculate the average production for a longer period of time?

An average for the global raspberry production is calculated by using the most recent 5 year data available (2016-2020). The global production for year 2019 stated in the abstract is replaced by this newly calculated average.

L127 (Table 1) The word „plant“ might be omitted from the title.

The word plant is omitted.

L135 “are” should be added in the sentence: In addition, plant-parasitic nematodes which are known to transmit plant viruses belong to the order of Dorylaimida and ARE limited to the families of Longidoridae and Trichodoridae [11].

The “are” is added as suggested.

L146 “circulative”, like propagative, should not be in parentheses.

The parentheses are removed.

L154 This is the first mention of RLSV. Therefore, the virus full name should be added. Iti s not clear if the complex encompasses three or four viruses since RLSV is written in parentheses.

The full name of the virus is added. The explanation for the virus is added with response to the next comment.

L160 The relation of RLMV and its isolate RLSV should be briefly clarified.

A brief explanation is added, explaining that RLMV and RLSV is later classified as isolate due to genetic similarities.

L186 colonies are instead of “colonies is”

This phrase is corrected.

L297 Thrips imaginis and T. fuscipennis are also not proven as vectors instead of “is proven as a vector”

This sentence is corrected.

L374 I suggest: The plant viruses associated with longidorid-nematodes are mainly nepoviruses. Otherwise, it should be: „mainly belong to genus Nepovirus

The phrase is corrected.

L443 a space should be added in “O. bruceata

The spacing is added.

Unless otherwise recommended by the journal, virus higher taxon name should be written in italics and begins with a capital letter (ICTV). This differs from the convention in botany and zoology, in which taxon names above the level of genus are not italicized.

The taxon name is check and corrected throughout the manuscript.

I would suggest adding one figure with photos of the most important raspberry virus vectors. It might be helpful for those virologists who are not very familiar with the topic.

We agreed that photos can be helpful, but unfortunately, we do not possess the original photos for all of these important virus vectors now. We only have two species of the aphids and we do not prefer to present a figure with incomplete collection of raspberry virus vectors. We wish for your understanding.

Reviewer 2 Report

The manuscript entitled “Raspberry viruses and their known and potential invertebrate vectors” is a well-written, interesting and comprehensive review on the present knowledge of the proven vectors and some other potential vectors of the 22 viruses that are known to infect raspberry.

In reviewing this manuscript, I have not found serious drawbacks. I only have some suggestions to the authors.

Given that the review focus is on knowledge and literature that associates several groups of invertebrates as proven or non-proven (potential) vectors for the transmission of raspberry viruses, and not on these viruses themselves (i.e. their life cycle, symptoms, epidemiology, etc., known raspberry viruses are merely listed), may be, a proper title for the manuscript should be “Known and potential invertebrate vectors of raspberry viruses”.

According to the abstract and conclusions, only some species of aphids and plant-parasitic nematodes are presented as proven vectors of raspberry viruses, while the eriophyid mite Phyllocoptes gracilis is presented as the suggested natural vector of raspberry leaf blotch virus  (the most probable, as indicated by strong evidence), in the absence of deeper knowledge on the transmission mechanism involved. However, the presentation of some information in this regard seems to be misleading: 1) this species (P. gracilis) is included in Table 2 as a known vector; 2) lines 447-448 (page 15) literally state that “Aphids, mites and nematodes are the only groups proven to be involved in raspberry virus transmission”. At least, an explanatory footnote in Table 2 or some explanatory comment in section 5.2 about the convenience of including pest management strategies for this probable mite vector species should be written.

Similarly, lines 211-213 (page 6) and the reference therein state that Macrosiphum euphorbiae, M. fragariae and Myzus ornatus “were found capable of transmitting several viruses of the RMD complex in Europe and North America”, which seems to point at these species as vectors of raspberry viruses. However, they are not included in Table 2 (instead, they are included in Tables 3 as potential vectors or non-vectors). Authors should provide a clearer criterion (or reference) to clarify the present status of these aphid species as known or potential vectors of raspberry viruses.  

In line 58 (page 2), authors chose the American use of the adjective “esophageal”, while later on (line 62) they chose the related word “oesophagus” as used in UK (instead of the American “esophagus”). They should unify the same regional use of these words.

Author Response

The manuscript entitled “Raspberry viruses and their known and potential invertebrate vectors” is a well-written, interesting and comprehensive review on the present knowledge of the proven vectors and some other potential vectors of the 22 viruses that are known to infect raspberry.

In reviewing this manuscript, I have not found serious drawbacks. I only have some suggestions to the authors.

First, the authors would like to thank the reviewer for taking the effort to carefully review the manuscript. All the suggestions provided are valuable and we appreciate them very much. Therefore, we have accepted all the recommendations and revised them as best we could.

Given that the review focus is on knowledge and literature that associates several groups of invertebrates as proven or non-proven (potential) vectors for the transmission of raspberry viruses, and not on these viruses themselves (i.e. their life cycle, symptoms, epidemiology, etc., known raspberry viruses are merely listed), may be, a proper title for the manuscript should be “Known and potential invertebrate vectors of raspberry viruses”.

We accepted the view of the reviewer, and decided to adopt this new title.

According to the abstract and conclusions, only some species of aphids and plant-parasitic nematodes are presented as proven vectors of raspberry viruses, while the eriophyid mite Phyllocoptes gracilis is presented as the suggested natural vector of raspberry leaf blotch virus  (the most probable, as indicated by strong evidence), in the absence of deeper knowledge on the transmission mechanism involved. However, the presentation of some information in this regard seems to be misleading:

1) this species (P. gracilis) is included in Table 2 as a known vector;

2) lines 447-448 (page 15) literally state that “Aphids, mites and nematodes are the only groups proven to be involved in raspberry virus transmission”. At least, an explanatory footnote in Table 2 or some explanatory comment in section 5.2 about the convenience of including pest management strategies for this probable mite vector species should be written.

Footnote is added to Table 2 to explain the inclusion of P. gracilis as a vector.

Similarly, lines 211-213 (page 6) and the reference therein state that Macrosiphum euphorbiaeM. fragariae and Myzus ornatus “were found capable of transmitting several viruses of the RMD complex in Europe and North America”, which seems to point at these species as vectors of raspberry viruses. However, they are not included in Table 2 (instead, they are included in Tables 3 as potential vectors or non-vectors). Authors should provide a clearer criterion (or reference) to clarify the present status of these aphid species as known or potential vectors of raspberry viruses.  

The three species are moved to Table 2 as known vectors.

In line 58 (page 2), authors chose the American use of the adjective “esophageal”, while later on (line 62) they chose the related word “oesophagus” as used in UK (instead of the American “esophagus”). They should unify the same regional use of these words.

The language consistency is checked and corrected accordingly.